# InfVSR: Breaking Length Limits of Generic Video Super-Resolution

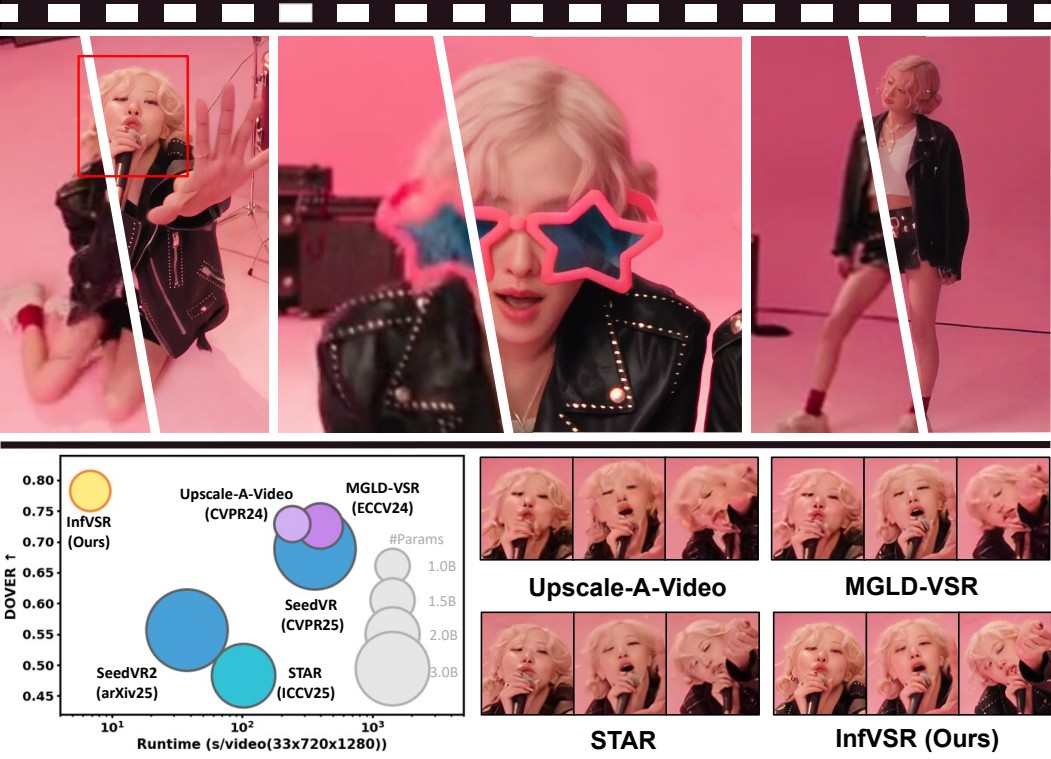

Figure 1: Speed and multi-frame comparisons. Our InfVSR is capable to seamlessly and streamingly upscale videos with **unbounded length**, and demonstrates the best quality and fastest speed among existing diffusion-based methods. Compared with MGLD-VSR (Yang et al., 2024), it is 58× faster.

## ABSTRACT

Real-world videos often extend over thousands of frames, posing unique demands far beyond current short benchmarks. Existing video super-resolution (VSR) approaches, however, face two persistent challenges when processing long sequences: (1) *Efficiency* due to the heavy cost of multi-step denoising for full-length sequences; and (2) *Scalability* hindered by temporal decomposition that causes artifacts and discontinuities. To break these limits, we propose **InfVSR**, which novelly reformulate VSR as an **autoregressive-one-step-diffusion** paradigm. This enables streaming inference while fully leveraging pre-trained video diffusion priors. *First*, we adapt the pre-trained DiT into a causal structure, maintaining both local and global coherence via rolling KV-cache and joint visual guidance. *Second*, we distill diffusion process into a single step efficiently, with patch-wise pixel supervision and cross-chunk distribution matching. Together, these designs enable efficient and scalable VSR for unbounded-length videos. To fill the gap in long-form video evaluation, we build a new benchmark tailored for extended sequences, and further introduce semantic-level metrics to comprehensively assess temporal consistency. Our method pushes the frontier of long-form VSR, achieves state-of-the-art quality with enhanced semantic consistency, and delivers up to **58×** speed-up over existing methods such as MGLD-VSR. Code will be released soon.

# 1 INTRODUCTION

The real world is witnessing an explosive growth in video content. Demand for high-quality video also surges. Video super-resolution (VSR), which aims to reconstruct high-resolution (HR) videos from low-resolution (LR) inputs, has thus emerged as a critical task. Among various approaches, generative models, particularly diffusion (Ho et al., 2020; Song et al., 2020)-based ones (Wang et al., 2025b), are most popular and preferred. With powerful pretrained priors, they can recover realistic details under complex real-world degradations (Wang et al., 2021; Chan et al., 2022b).

However, despite recent progress in generation quality (Xie et al., 2025; Wang et al., 2025b; Chen et al., 2025c; Wang et al., 2025c;a), there remains a key obstacle to their practical deployment: they are limited to short video clips, while real world videos often extend over thousands of frames. In terms of *efficiency*, super-resolving 500 frame 720p video using Upscale-A-Video (Yang et al., 2024) requires over 1 hour. It is even more intolerable when users can only view the result after the entire video has been processed. In terms of *scalability*, methods like SeedVR (Wang et al., 2025b) runs easily out of memory on an A800 80GB GPU when processing over 100 frame sequences. Temporal decomposition, though necessary, breaks the fine-grained temporal modeling originally learned. Although progressive patch aggregation (Wang et al., 2024a; 2025c) can smooth some transitions, they often introduce semantic inconsistencies and artifacts (Du et al., 2025). Both weaknesses pose an urgent need for a framework that fundamentally breaks the length limits.

In the first exploration of such framework, we propose **InfVSR**, which novelly reformulate VSR as an **autoregressive-one-step-diffusion** (AR-OSD) paradigm. Based on observation that powerful T2V diffusion models (Yang et al., 2025; Wan Team et al., 2025) boost recent VSR quality breakthoughs, our core idea is to maximally retain the capabilities of pre-trained T2V priors, while introducing a new autoregressive formulation for temporal modeling. To be specific, we partition the video into non-overlapping temporal chunks and apply (1) **intra-chunk diffusion**, where the generative prior is invoked to refine the current chunk, and (2) **inter-chunk autoregression**, where temporal consistency is preserved by propagating information from past chunks. This paradigm well supports seamless streaming, as explored in prior video generation work (Zhang & Agrawala, 2025; Sánchez et al., 2025; Huang et al., 2025; Chen et al., 2025b; Teng et al., 2025). We go further by extending this to high-resolution restoration, achieving superior quality with only a single-step diffusion process.

To support AR inference, the pre-trained DiT backbone needs to be converted into a causal architecture. We thus design a dual-temporal modeling strategy tailored for VSR. For **local smoothness**, we retain a rolling KV-cache in the self-attention module. This directly propagates high-resolution, adjacent context, preserves the sequence modeling capacity of DiT, and keeps low computational overhead. For **global coherence**, we construct a joint visual guidance. Visual prompt is extracted from LR reference frames, projected into cross-attention, and shared across all chunks. This compensates for the information loss caused by cache truncation, and maintains global style and identity guidance. With this design, our method only attends to a local sequence at each denoising step, yet preserves smooth transitions and strong temporal consistency. This enables scalable inference over arbitrarily long videos, with constant memory consumption over time.

Training such a framework is particularly challenging. We need to maintain high quality, fidelity and consistency under a single-step diffusion setting, while also alleviating the memory burden from the decoder, repeated forward passes, and additional teacher modules. To this end, we introduce two complementary objectives. A **patch-wise pixel supervision** guides detail reconstruction with significantly reduced memory, while preserving training on high resolution through random spatial sampling. A **cross-chunk distribution matching** (Sun & et al., 2024; Yin et al., 2024) constrains long-range temporal dynamics by aligning the output distribution of several autoregressive steps with what a teacher model learns. In addition, a **two-stage** schedule first adapts the model to one-step HR prediction, then introduces autoregression with cache and prompts. Notably, our model is trained entirely on short videos, yet generalizes to arbitrary lengths at test time. Also, with such a training strategy, our model achieves state-of-the-art fidelity, visual quality and temporal consistency across multiple datasets, using only a single diffusion step. It opens new possibilities for long-range video enhancement and lays the foundation for practical deployment of generic VSR systems.

Moreover, existing benchmarks (Tao et al., 2017; Yi et al., 2019; Zhou et al., 2024; Chan et al., 2022b) remain focused on short videos, typically no more than 100 frames, and offer limited evaluation of high-level consistency. To fill this gap, we introduce MovieLQ, a benchmark of 1000-frame-long, real-world degraded videos. We further adopt semantic-level metrics from VBench (Huang et al.,

2024), enabling a more comprehensive evaluation of long-range stability and perceptual coherence. Our contributions can be summarized as follows:

- We propose **InfVSR**, the first T2V-based autoregressive-one-step-diffusion framework for real-world VSR, which supports ultra-efficient inference on unbounded-length sequences.
- For AR framework, we introduce a dual-timescale mechanism with rolling KV-cache and joint visual guidance. For training, we design patch-wise pixel supervision and cross chunk distribution matching, well balancing fidelity, consistency, and efficiency.
- We establish MovieLQ, a benchmark tailored for long-form VSR under real-world degradations, and first adopt metrics assessing semantic-level temporal consistency.
- Extensive experiments demonstrate the effectiveness and efficiency of our method. We achieve SOTA performance with significantly reduced latency and memory cost.

## 2 RELATED WORKS

**Video Super-Resolution.** Video super-resolution aims to recover high-quality videos from degraded inputs. Early methods rely on RNNs (Chan et al., 2021; 2022a) or window-based transformers (Li et al., 2020; Yi et al., 2019) trained on synthetic degradation, which limits their performance in real-world scenarios. Later works (Chan et al., 2022b; Yang et al., 2021) introduce mixed degradations such as blur and compression, but still struggle to produce sharp and stable results. Diffusion models (Song et al., 2020; Ho et al., 2020; Rombach et al., 2022) have recently brought a major shift by introducing strong generative priors. Text-to-Image (T2I) based methods (Rombach et al., 2022; Peebles & Xie, 2023; Esser et al., 2024), such as Upscale-A-Video (Zhou et al., 2024) and MGLD-VSR (Yang et al., 2024), inject optical flow (Ranjan & Black, 2017; Teed & Deng, 2020) or temporal modules into pretrained image diffusion backbones. However, their frame alignment remains fragile and error-prone. Text-to-Video (T2V) (Blattmann et al., 2023; Yang et al., 2025; Wan Team et al., 2025; HaCohen et al., 2024) based methods like STAR (Xie et al., 2025) and SeedVR (Wang et al., 2025b) leverage video-scale priors and achieve significantly better temporal coherence. Models such as DOVE (Chen et al., 2025c) and SeedVR2 (Wang et al., 2025a) further distill multi-step denoising into one-step inference, offering large speedups. Yet these models still suffer from memory cost scaling with video length and temporal inconsistency caused by independent chunk inference. We address both limitations by reformulating VSR as an autoregressive-one-step-diffusion paradigm, while simultaneously leveraging strong video generative priors.

**Autoregressive Video Generation.** Recent progress in large-scale T2V pretraining (Yang et al., 2025; Kong et al., 2024; Wan Team et al., 2025) has paved the way for autoregressive video generation, which in turn emphasizes real-time operation and stronger controllability. One line (Kondratyuk et al., 2024; Yuan et al., 2025) adopts GPT-style next-token prediction in latent space. Another (Sánchez et al., 2025; Teng et al., 2025; Chen et al., 2025b; Zhang & Agrawala, 2025) explores AR-diffusion by performing intra-frame denoising and inter-frame rollout, which better matches video dynamics and uses T2V priors. Training strategies include diffusion-forcing (Chen et al., 2025a), teacher-forcing (Williams & Zipser, 1989), and self-forcing (Huang et al., 2025). Our work is, to our knowledge, the first to explore AR-diffusion models for long-form VSR. And beyond simply applying existing generation techniques, we tailor the design to VSR's LR inputs, high-fidelity demands, and high-resolution training. Also notably, we achieve inference with one forward pass, without multiple sampling steps or an additional pass for clean KV cache.

**One-step Diffusion.** One-step diffusion has emerged as a key solution to accelerate diffusion-based generation. Existing approaches include rectified flow matching (Liu et al., 2022; 2023), score distillation (*e.g.*, VSD (Wang et al., 2023c), DMD (Sun & et al., 2024; Yin et al., 2024)), and post-hoc fine-tuning via regression (Zhang et al., 2018; Ding et al., 2020) or adversarial training (Lin et al., 2025). In ISR, these methods have been widely adopted, such as direct sampling in SinSR (Wang et al., 2024b), VSD (Wang et al., 2023c) in OSEDiff (Wu et al., 2024a), and target score distillation in TSD-SR (Dong et al., 2025). In VSR, one-step generation has just started to emerge. UltraVSR (Liu et al., 2025) and DLoraL (Sun et al., 2025) extend ISR-based one-step models with temporal alignment. DOVE (Chen et al., 2025c) replaces multi-step denoising with a direct regression loss (Ding et al., 2020). SeedVR2 (Wang et al., 2025a) uses APT (Lin et al., 2025) to match output distribution with a pretrained prior. While these designs significantly accelerate inference, they still require full-sequence processing, making them hard to scale to long videos. In contrast, we adopt an AR-OSD paradigm, combining the speed of single-step inference with scalability of autoregressive modeling.

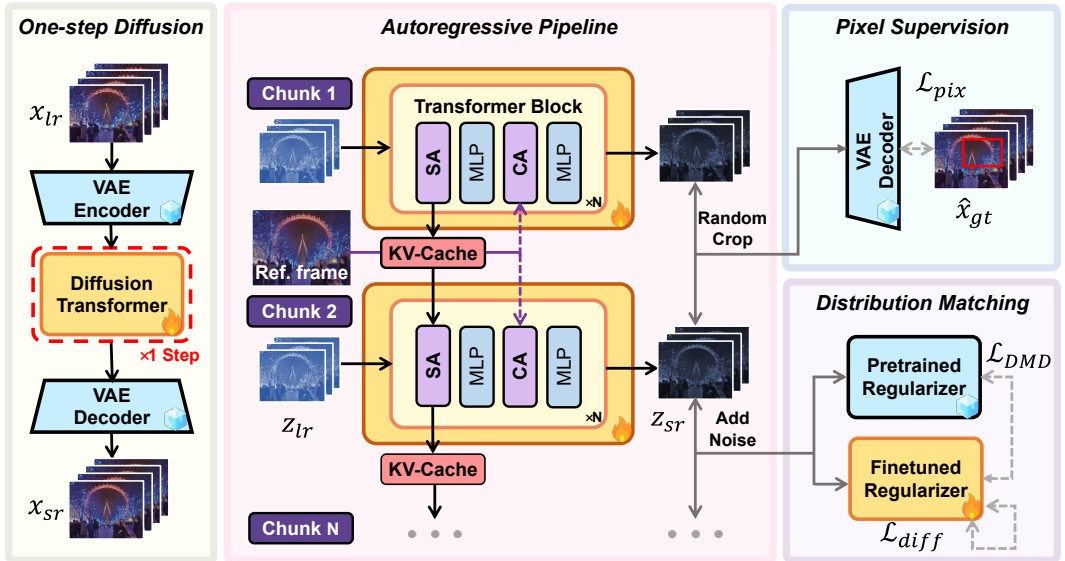

Figure 2: Overview of the framework and training strategy of InfVSR. Our method combines intra-chunk **one-step diffusion** with inter-chunk **autoregression** for efficient and scalable VSR. AR is supported by local KV-cache and global joint visual guidance. To enable effective and efficient training, we adopt two objectives: (1) **patch-wise pixel supervision** which guides detail reconstruction with significantly reduced memory decoding through random spatial cropping; and (2) **cross-chunk distribution matching** which enforces high-level consistency with a pretrained and a finetuned regularizer, following (Wang et al., 2023c; Sun & et al., 2024; Wu et al., 2024a).

## 3 METHODOLOGY

### 3.1 PROBLEM FORMULATION

In this work, we make the first attempt to reformulate VSR as an **autoregressive-one-step-diffusion** (AR-OSD) paradigm. Our core idea is to maximally retain the capabilities of pre-trained T2V priors, while introducing a new autoregressive formulation for temporal modeling. Specifically, as illustrated in Fig. 2, we divide the video into non-overlapping chunks and model: (1) **intra-chunk diffusion**, where the generative prior is invoked to refine the current chunk, and (2) **inter-chunk autoregression**, where temporal consistency is preserved by propagating information from past chunks. Formally, let the video be divided into $K$ non-overlapping chunks. For each chunk $k$, let $\mathbf{x}_k$ and $\mathbf{y}_k$ denote the LR input and the HR output, respectively. We factorize the joint distribution autoregressively as:

$$p(\mathbf{y}_{1:K} \mid \mathbf{x}_{1:K}) = \prod_{k=1}^{K} p(\mathbf{y}_k \mid \mathbf{x}_k, \mathcal{P}_k), \qquad (1)$$

where $\mathcal{P}_k$ represents the autoregressive context collected from previously generated chunks. Each conditional $p(\mathbf{y}_k \mid \mathbf{x}_k, \mathcal{P}_k)$ is approximated by a one-step diffusion mapping from $\mathbf{x}_k$ to $\mathbf{y}_k$:

$$\mathbf{y}_k = G_\theta(\mathbf{x}_k, \mathcal{P}_k). \qquad (2)$$

Here, $G_\theta$ is a generator adapted from a pre-trained T2V diffusion backbone, conditioned on the current LR chunk and a compact context $\mathcal{P}_k$ derived from past outputs.

### 3.2 CAUSAL DiT ARCHITECTURE

Our method builds upon a pretrained T2V diffusion model, which typically consists of: (1) a 3D VAE that compresses spatial dimensions by 8× and temporal length from 4n+1 to n+1 frames, and (2) a pre-trained 3D DiT. As this DiT is originally pre-trained for generating short video clips with full attention, adaptions need to be made to support causal inference. To be specific, we introduce a dual-timescale temporal modeling based on VSR specialties.

**Local Smoothness via Rolling KV-cache.** The pretrained DiT exhibits strong sequence modeling capabilities over tokenized video representations. To enhance scalability for long-form inference, we draw inspiration from causal inference in large language models and employ a KV-cache mechanism. Specifically, in the self-attention layers, we assign absolute positional embeddings to both queries and keys based on the chunk's position in the original video timeline. The key and value tensors

accumulated from previous chunks are then concatenated with the current chunk's KV representations. This design enables the model to maintain temporal awareness across chunk boundaries without reprocessing earlier frames. As a result, the model achieves smooth transitions and strong local temporal consistency, while keeping memory usage constant over time.

Unlike video generation where a long KV cache is typically required to preserve motion continuity, VSR does not necessitate such extensive memory. The LR input already provides strong structural priors. We argue that maintaining a fixed-length, rolling KV cache brings two key benefits in the VSR setting: (1) it prevents memory and computation from growing unbounded with video length; (2) it ensures that each chunk during training and inference faces a more consistent local distribution, which facilitates better convergence and generalization. Therefore, we adopt a rolling-update strategy for the KV cache, where only a limited number of past frames are used for the current chunk's attention.

**Global coherence via Joint Visual Guidance.** While the rolling KV cache preserves local temporal continuity with low computational overhead, cache truncation may cause information loss. In standard video generation settings, this often necessitates external memory banks or identity supervision to maintain long-term consistency. In the VSR setting, however, the low-resolution input itself provides a strong global prior. We leverage this property by treating the LR video as a persistent global reference throughout restoration. Specifically, we select scene-representative keyframes and encode them using a pretrained visual encoder (i.e. DAPE (Wu et al., 2024b)). The visual embeddings are injected as prompts into the cross-attention layers of the DiT backbone. These prompts remain constant across all chunks, providing a consistent global anchor for identity and context.

By combining rolling KV-cache with joint visual guidance, our framework achieves both local smoothness and global coherence across arbitrarily long sequences, with low computational overhead.

### 3.3 EFFICIENT AUTOREGRESSIVE POST-TRAINING

Training our AR-OSD VSR model requires careful design to balance fidelity, temporal consistency, and memory efficiency. We design a lightweight yet effective training scheme composed of two supervision losses as illustrated in Fig. 2 and a staged curriculum.

**Patch-wise Pixel Supervision.** A pixel-domain reconstruction loss is widely adopted in one-step ISR methods (Wu et al., 2024a; Dong et al., 2025), for recovering high-fidelity visual details. However, in VSR, the $8\times$ spatial and $4\times$ temporal upsampling in the 3D VAE decoder incurs prohibitive memory overhead, which rapidly increases with video resolution. It hinders training DiT at high resolutions, making it difficult to handle long token sequences especially crucial for VSR tasks.

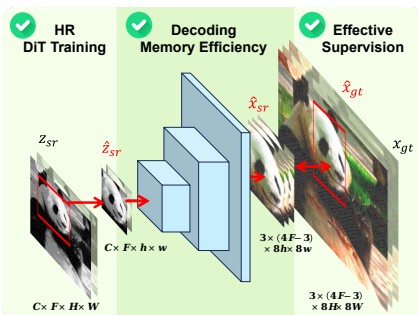

Figure 3: Illustration of our pixel loss.

To address this, we propose **patch-wise pixel supervision**, which is built upon the insight that, in the spatial domain, applying reconstruction loss to randomly cropped video patches is **expectationally equivalent** to computing the loss over the full video. Moreover, it's based on the VAE decoder's ability to reconstruct small spatial tiles almost identically to decoding the entire frame, thanks to the locality of CNNs and the officially supported tiling mechanism. Specifically, let the predicted latent video $\hat{\mathbf{z}} \in \mathbb{R}^{B \times C \times F \times H \times W}$, and let $D(\cdot)$ be the 3D VAE decoder. We define a random spatial cropping operator $\mathcal{C}_{\text{lat}}(\cdot)$ that extracts a patch of size $h \times w$ from the latent tensor, and another operator $\mathcal{C}_{\text{pix}}(\cdot)$ for the HR video, which extracts the corresponding patch in ground-truth space. Due to the $8\times$ spatial upsampling in $D(\cdot)$, we align the cropping windows such that: $\mathcal{C}_{\text{lat}}$ samples a random crop window at position $(i, j)$ in latent space, and $\mathcal{C}_{\text{pix}}$ applies the same window at position $(8i, 8j)$ in pixel space. Then, the decoded high-resolution patch sequence is:

$$\hat{\mathbf{x}}_{\text{sr}} = D\left(\mathcal{C}_{\text{lat}}(\hat{\mathbf{z}})\right), \quad \hat{\mathbf{x}}_{\text{gt}} = \mathcal{C}_{\text{pix}}(\mathbf{x}_{\text{gt}}), \tag{3}$$

where $\hat{\mathbf{x}}_{\text{sr}}, \hat{\mathbf{x}}_{\text{gt}} \in \mathbb{R}^{B \times 3 \times (4F-3) \times 8h \times 8w}$ are the decoded super-resolved and ground-truth patch sequences, respectively. We then apply two types of loss functions over these cropped patches. The first is a fidelity loss that encourages spatial accuracy and structural realism:

$$\mathcal{L}_{\text{fidel}} = \lambda_{\text{mse}} \cdot \mathcal{L}_{\text{mse}}(\hat{\mathbf{x}}_{\text{sr}}, \hat{\mathbf{x}}_{\text{gt}}) + \lambda_{\text{dists}} \cdot \mathcal{L}_{\text{dists}}(\hat{\mathbf{x}}_{\text{sr}}, \hat{\mathbf{x}}_{\text{gt}}), \tag{4}$$

where $\mathcal{L}_{\text{mse}}$ and $\mathcal{L}_{\text{dists}}$ denote the mean squared error and DISTS (Ding et al., 2020) perceptual loss respectively, and $\lambda_{\text{mse}}$ and $\lambda_{\text{dists}}$ are loss scalers. In addition, we introduce a temporal smoothness

loss to enforce local temporal consistency by aligning frame-wise differences:

$$\mathcal{L}_{\text{temp}} = \lambda_{\text{temp}} \cdot \left\| (\hat{\mathbf{x}}_{\text{gt}}^{t+1} - \hat{\mathbf{x}}_{\text{gt}}^{t}) - (\hat{\mathbf{x}}_{\text{sr}}^{t+1} - \hat{\mathbf{x}}_{\text{sr}}^{t}) \right\|_2^2, \tag{5}$$

where $\hat{\mathbf{x}}_{\text{gt}}^{t}$ and $\hat{\mathbf{x}}_{\text{sr}}^{t}$ denote the ground-truth and predicted frames at frame $t$ respectively, and $\lambda_{\text{temp}}$ is the loss scaler. This loss constrains temporal variations instead of pixel-wise alignment, making it suited to small-patch windows. The total patch-wise supervision loss is $\mathcal{L}_{\text{pix}} = \mathcal{L}_{\text{fidel}} + \mathcal{L}_{\text{temp}}$.

**Cross-Chunk Distribution Matching.** While the temporal smoothness loss $\mathcal{L}_{\text{temp}}$ encourages frame-to-frame consistency, we observe identity inconsistency and temporal incoherence across chunk boundaries, suggesting that temporal supervision should extend beyond adjacent frames. To address this, we introduce a **cross-chunk distribution matching** loss $\mathcal{L}_{\text{DMD}}$, which aligns the high-level feature distribution of generated videos with that of real videos over a longer temporal range. Specifically, we apply $\mathcal{L}_{\text{DMD}}$ to sequences concatenating three adjacent autoregressively generated chunks. We minimize the KL divergence between the distributions of generated features and ground-truth features extracted from a frozen teacher video model:

$$\nabla_\phi \mathcal{L}_{\text{DMD}} = \mathbb{E}_t \left( \nabla_\phi \, \text{KL} \left( p_{\text{gen}} \, \| \, p_{\text{data}} \right) \right), \tag{6}$$

where $p_{\text{gen}}$ denotes the distribution of features from the generated triplet-chunk sequence, and $p_{\text{data}}$ is the corresponding real distribution learned from the teacher model. This KL-based regularization improves long-term semantic consistency and mitigates identity drift across chunks.

**Two-Stage Curriculum Training.** AR training is expensive as each update requires multiple forward passes and additional teacher modules. In VSR, decoding and training at high resolution makes this burden prohibitive. We therefore adopt a two-stage curriculum. **Stage I: Initialization** optimizes only $\mathcal{L}_{\text{pix}}$ on high-resolution, long clips to fit one-step diffusion, while **Stage II: AR Adaptation** trains at lower resolution with KV cache and $\mathcal{L}_{\text{DMD}}$ to adapt to autoregressive inference. Empirically, Stage I already yields strong VSR quality, and Stage II converges rapidly to an AR-ready model, allowing us to train the full system at substantially lower cost.

### 3.4 MovieLQ Dataset and Benchmark

Existing VSR benchmarks, such as UDM10 (Tao et al., 2017), SPMCS (Yi et al., 2019), YouHQ (Zhou et al., 2024), and VideoLQ (Chan et al., 2022b), typically consist of short clips under 100 frames. While suitable for evaluating base quality, they fail to reflect the real-world constraints of long-form video enhancement, where current models must operate in a memory-efficient, segment-wise manner. To bridge this gap, we introduce **MovieLQ**, a new benchmark comprising 1000-frame-long, single-shot videos sourced from various video hosting sites such as Vimeo and pixabay with Creative Commons license. Following (Chan et al., 2022b), all clips exhibit real-world degradation patterns without synthetic corruption. To enable fair comparison, we apply progressive patch aggregation following (Wang et al., 2024a; 2025c) to current memory-intensive methods.

## 4 Experiments

### 4.1 Experimental Settings

**Datasets.** Our method is data-efficient. For training, we only use the REDS (Nah et al., 2019) dataset and segment the videos into approximately 1K clips. We adopt the RealBasicVSR (Chan et al., 2022b) degradation pipeline. For evaluation, we apply both synthetic and real-world datasets. The synthetic datasets include UDM10 (Tao et al., 2017) and SPMCS (Yi et al., 2019), using the same degradations as training. For real-world datasets, we apply MVSR4x (Wang et al., 2023b), VideoLQ (Chan et al., 2022b), and our proposed MovieLQ. All experiments are conducted with a scaling factor $\times 4$.

**Evaluation Metrics.** We adopt multiple evaluation metrics to comprehensively assess fidelity, perceptual quality, and temporal consistency. For fidelity, we employ full-reference image quality assessment (IQA) metrics including PSNR, SSIM (Wang et al., 2004), LPIPS (Zhang et al., 2018), and DISTS (Ding et al., 2020). For perceptual quality, we adopt no-reference IQA metrics like MUSIQ (Ke et al., 2021) and CLIPIQA (Wang et al., 2023a), and take DOVER (Wu et al., 2023) for video quality assessment. For temporal consistency, we adopt the flow warping error $E_{warp}^*$ ($\times 10^{-3}$) (Lai et al., 2018) to assess pixel-level consistency. Moreover, we take background consistency (BC), subject consistency (SC), and motion smoothness (MS) from VBench (Huang et al., 2024) to assess temporal consistency at the semantic level. With these metrics, a comprehensive evaluation is conducted.

**Implementation Details.** Our InfVSR is built upon the T2V model Wan 2.1 (1.3B) (Wan Team et al., 2025) and is trained on 4 NVIDIA A800-80G GPUs with AdamW (Loshchilov & Hutter, 2018). We use gradient accumulation for larger batch size. We set the cache and chunk length both to 3. In Stage

Table 1: Quantitative comparison with state-of-the-art methods. The best and second performances are marked in red and orange respectively. Our method outperforms on various datasets and metrics.

| Datasets | Metrics | RealBasicVSR CVPR 2022 | RealViFormer ECCV 2024 | Upscale-A-Video CVPR 2024 | MGLD-VSR ECCV 2024 | STAR ICCV 2025 | SeedVR CVPR 2025 | SeedVR2 arXiv 2025 | Ours - |
|---|---|---|---|---|---|---|---|---|---|
| UDM10 | PSNR ↑ | 24.13 | 24.64 | 21.72 | 24.23 | 23.47 | 23.39 | 25.38 | 24.86 |
| | SSIM ↑ | 0.6801 | 0.6947 | 0.5913 | 0.6957 | 0.6804 | 0.6843 | 0.7764 | 0.7274 |
| | LPIPS ↓ | 0.3908 | 0.3681 | 0.4116 | 0.3272 | 0.4242 | 0.3583 | 0.2868 | 0.2972 |
| | DISTS ↓ | 0.2067 | 0.2039 | 0.2230 | 0.1677 | 0.2156 | 0.1339 | 0.1512 | 0.1422 |
| | MUSIQ ↑ | 59.06 | 57.90 | 59.91 | 60.55 | 41.98 | 53.62 | 49.95 | 62.88 |
| | CLIP-IQA ↑ | 0.3494 | 0.4157 | 0.4697 | 0.4557 | 0.2417 | 0.3145 | 0.2987 | 0.5142 |
| | DOVER ↑ | 0.7564 | 0.7303 | 0.7291 | 0.7264 | 0.4830 | 0.6889 | 0.5568 | 0.7826 |
| | $E^*_{warp}$ ↓ | 3.10 | 2.29 | 3.97 | 3.59 | 2.08 | 3.24 | 1.98 | 1.95 |
| SPMCS | PSNR ↑ | 22.17 | 22.72 | 18.81 | 22.39 | 21.24 | 21.22 | 22.57 | 22.25 |
| | SSIM ↑ | 0.5638 | 0.5930 | 0.4113 | 0.5896 | 0.5441 | 0.5672 | 0.6260 | 0.5697 |
| | LPIPS ↓ | 0.3662 | 0.3376 | 0.4468 | 0.3262 | 0.5257 | 0.3488 | 0.3176 | 0.3166 |
| | DISTS ↓ | 0.2164 | 0.2108 | 0.2452 | 0.1960 | 0.2872 | 0.1611 | 0.1757 | 0.1742 |
| | MUSIQ ↑ | 66.87 | 64.47 | 69.55 | 65.56 | 36.66 | 62.59 | 60.17 | 67.75 |
| | CLIP-IQA ↑ | 0.3513 | 0.4110 | 0.5248 | 0.4348 | 0.2646 | 0.3945 | 0.3811 | 0.5319 |
| | DOVER ↑ | 0.6753 | 0.5905 | 0.7171 | 0.6754 | 0.3204 | 0.6576 | 0.6320 | 0.7302 |
| | $E^*_{warp}$ ↓ | 1.88 | 1.46 | 4.22 | 1.68 | 1.01 | 1.72 | 1.23 | 1.25 |
| MVSR4x | PSNR ↑ | 21.80 | 22.44 | 20.42 | 22.77 | 22.42 | 21.54 | 21.88 | 22.49 |
| | SSIM ↑ | 0.7045 | 0.7190 | 0.6117 | 0.7417 | 0.7421 | 0.6869 | 0.7678 | 0.7373 |
| | LPIPS ↓ | 0.4235 | 0.3997 | 0.4717 | 0.3568 | 0.4311 | 0.4944 | 0.3615 | 0.3452 |
| | DISTS ↓ | 0.2498 | 0.2453 | 0.2673 | 0.2245 | 0.2714 | 0.2229 | 0.2141 | 0.2107 |
| | MUSIQ ↑ | 62.96 | 61.99 | 69.80 | 53.46 | 32.24 | 42.56 | 35.29 | 64.03 |
| | CLIP-IQA ↑ | 0.4118 | 0.5206 | 0.6106 | 0.3769 | 0.2674 | 0.2272 | 0.2371 | 0.5229 |
| | DOVER ↑ | 0.6846 | 0.6451 | 0.7221 | 0.6214 | 0.2137 | 0.3548 | 0.3098 | 0.6872 |
| | $E^*_{warp}$ ↓ | 1.69 | 1.25 | 5.07 | 1.55 | 0.61 | 2.73 | 1.08 | 1.03 |
| VideoLQ | MUSIQ ↑ | 55.62 | 52.18 | 55.04 | 51.00 | 39.66 | 54.41 | 39.10 | 56.26 |
| | CLIP-IQA ↑ | 0.3433 | 0.3553 | 0.4132 | 0.3465 | 0.2652 | 0.3710 | 0.2359 | 0.4454 |
| | DOVER ↑ | 0.7388 | 0.6955 | 0.7370 | 0.7421 | 0.7080 | 0.7435 | 0.6799 | 0.7556 |
| | $E^*_{warp}$ ↓ | 5.97 | 4.47 | 13.47 | 6.79 | 5.96 | 9.27 | 8.34 | 7.52 |
| MovieLQ | MUSIQ ↑ | 62.59 | 63.74 | 68.49 | 67.90 | 56.57 | 64.42 | 61.13 | 68.65 |
| | CLIP-IQA ↑ | 0.4672 | 0.4227 | 0.5117 | 0.5591 | 0.3411 | 0.505 | 0.4468 | 0.5888 |
| | DOVER ↑ | 0.8234 | 0.8273 | 0.775 | 0.8402 | 0.7565 | 0.8145 | 0.8031 | 0.8447 |
| | $E^*_{warp}$ ↓ | 3.39 | 2.24 | 5.53 | 3.67 | 3.11 | 4.70 | 4.26 | 2.88 |

1, we use a resolution of $33 \times 720 \times 1280$ with a window size of $320 \times 480$, a batch size of 8, and a learning rate of $5 \times 10^{-5}$. In Stage 2, the resolution is cropped to $33 \times 480 \times 720$ with a window size of $160 \times 160$, a batch size of 32, and a learning rate of $1 \times 10^{-5}$. We take $\lambda_{mse} = \lambda_{dists} = \lambda_{temp} = 1$.

## 4.2 COMPARISON WITH STATE-OF-THE-ART METHODS

To verify the effectiveness of our approach, we compare our InfVSR with recent state-of-the-art VSR methods: RealBasicVSR (Chan et al., 2022b), RealViFormer (Zhang & Yao, 2024), Upscale-A-Video (Zhou et al., 2024), MGLD-VSR (Yang et al., 2024), STAR (Xie et al., 2025), SeedVR (Wang et al., 2025b), and SeedVR2 (Wang et al., 2025a).

**Quantitative Results.** As shown in Table 1, our InfVSR achieves outstanding performance across diverse datasets. For fidelity, our method consistently ranks first or second across most benchmarks. For perceptual quality, our model achieves top-1 scores on five datasets, demonstrating its strong ability to generate visually pleasing results. In terms of temporal consistency (i.e., $E^*_{warp}$), our approach attains the lowest errors on UDM10 and performs robustly on other sets. Overall, these results verify the effectiveness of our model in producing high-fidelity, perceptually natural, and temporally stable videos across both synthetic and real-world scenarios.

**Qualitative Results.** We provide visual comparisons on both synthetic (i.e., SPMCS) and real-world (i.e., VideoLQ) videos in Fig. 4. Our InfVSR can handle complex degradations and produce more realistic results. For instance, in the first case, InfVSR successfully reconstructs the building's structure and textures under extremely severe degradation, whereas the others appear blurry or distorted. Similarly, our method restores clear text edges in the second example. More visual results are provided in the supplementary material.

**Temporal Consistency.** Given the importance of temporal consistency in VSR, we evaluate it from both pixel-level and semantic-level perspectives. We visualize the temporal profile in Fig. 5, where our method exhibits significantly smoother and

Table 2: Vbench Results on UDM10 and MovieLQ.

| Method | UDM10 | | | MovieLQ | | |
|---|---|---|---|---|---|---|
| | SC | BC | MS | SC | BC | MS |
| UAV | 0.9496 | 0.9489 | 0.9849 | 0.9494 | 0.9456 | 0.9749 |
| MGLD | 0.9413 | 0.9455 | 0.9863 | 0.9432 | 0.9434 | 0.9875 |
| STAR | 0.9450 | 0.9520 | 0.9899 | 0.9546 | 0.9532 | 0.9873 |
| SeedVR | 0.9625 | 0.9536 | 0.9844 | 0.9510 | 0.9405 | 0.9859 |
| Ours | 0.9632 | 0.9523 | 0.9910 | 0.9593 | 0.9513 | 0.9886 |

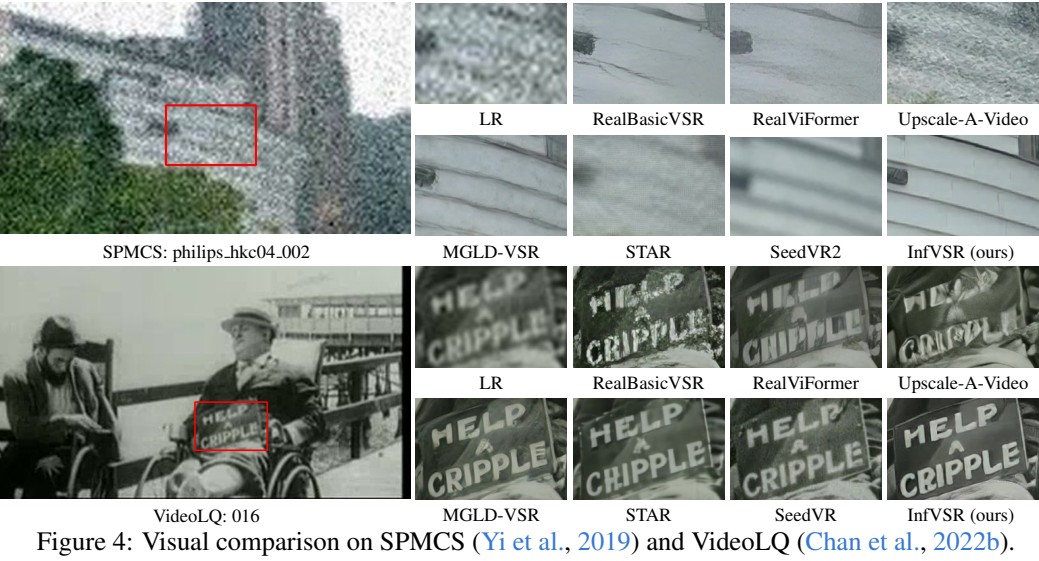

Figure 4: Visual comparison on SPMCS (Yi et al., 2019) and VideoLQ (Chan et al., 2022b).

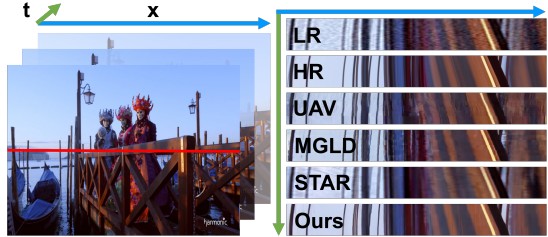

Figure 5: Comparison of temporal profile (stacking the red line across frames).

| Method | 33×720p Time | Mem | 100×720p Time | Mem |
|---|---|---|---|---|
| UAV-s30 | 241.43 | 43.38 | 731.60 | 43.38 |
| MGLD-s50 | 396.06 | 27.70 | 1,200.20 | 27.70 |
| STAR-s15 | 101.59 | 22.14 | 314.84 | 52.99 |
| SeedVR-s50 | 360.66 | 70.44 | 893.03 | 72.44 |
| SeedVR2-s1 | 37.43 | 61.13 | 68.18 | 61.44 |
| Ours-s1 | **6.82** | **20.39** | **20.70** | **20.39** |

Table 3: Comparison of running time of different diffusion-based methods.

more coherent transitions over time. Results for VBench metrics are presented in Tab. 2 , where our method achieves top-1 or top-2 performance both under full-temporal settings (e.g., UDM10) and progressive patch aggregation pipelines (e.g., MovieLQ). These results highlight the strength of our framework in preserving semantic consistency across long-range sequences. Fig. 6 also illustrates our consistency in identity preserving across chunks.

**Efficiency.** We compare our method against both multi-step and single-step baselines in terms of runtime and memory usage in Tab. 3. All experiments are conducted on a single NVIDIA A800-80G GPU for fair comparison. On 720p videos with 33 frames, our method takes only 6.82 seconds, which is 58× faster than MGLD-VSR (Yang et al., 2024) and 5.48× faster than recent prevailing one-step method, SeedVR2 (Wang et al., 2025a). For long videos (e.g., 100 frames), our memory usage remains constant, and runtime grows linearly with respect to the input length, while still being significantly faster than existing methods. These results demonstrate our high efficiency.

### 4.3 ABLATION STUDY

We conduct ablations on our effectiveness of our designs and settings. All training configurations are kept consistent with settings described in Sec. 4.1 and all experiments are conducted on UDM10 (Tao et al., 2017). Results are presented in Tab. 4.

**Effectiveness of AR inference.** We verify the effectiveness of our AR inference strategy in Tab. 4a and the right 3 columns in Fig. 6. Compared to simple chunking without cache, AR significantly improves both perceptual quality and temporal consistency, as the latter suffers from small receptive fields and lacks long-range temporal modeling. Although adding overlap and blending (Aggregation) in chunking-based methods improves pixel-level smoothness (as reflected by lower $E^*_{warp}$), it fails to enhance semantic-level consistency. In contrast, our AR design leverages pretrained T2V priors more effectively, leading to better semantic alignment and consistency across long videos.

**Effectiveness of joint guidance.** We study the role of joint guidance during inference in Tab. 4b. Without guidance, generation quality degrades across all metrics. Extracting separate guidance for each chunk introduces inconsistency across segments, resulting in slightly worse performance in

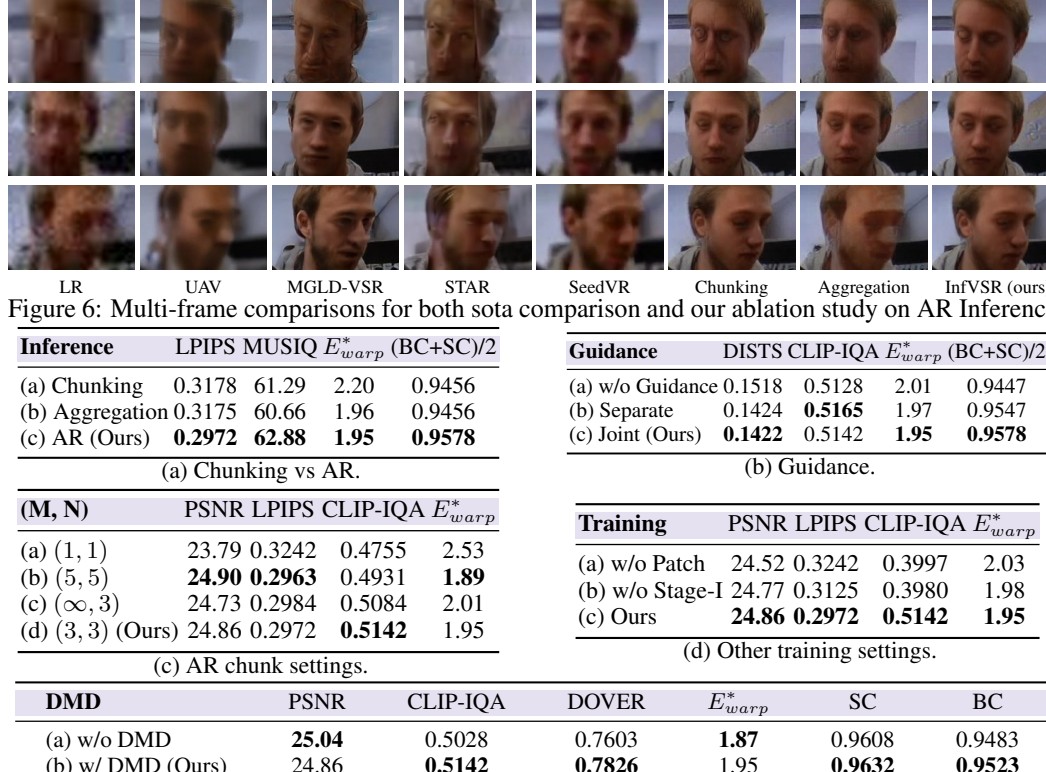

LR  UAV  MGLD-VSR  STAR  SeedVR  Chunking  Aggregation  InfVSR (ours)

Figure 6: Multi-frame comparisons for both sota comparison and our ablation study on AR Inference.

| Inference | LPIPS | MUSIQ | $E^*_{warp}$ | (BC+SC)/2 |
|---|---|---|---|---|
| (a) Chunking | 0.3178 | 61.29 | 2.20 | 0.9456 |
| (b) Aggregation | 0.3175 | 60.66 | 1.96 | 0.9456 |
| (c) AR (Ours) | **0.2972** | **62.88** | **1.95** | **0.9578** |

(a) Chunking vs AR.

| Guidance | DISTS | CLIP-IQA | $E^*_{warp}$ | (BC+SC)/2 |
|---|---|---|---|---|
| (a) w/o Guidance | 0.1518 | 0.5128 | 2.01 | 0.9447 |
| (b) Separate | 0.1424 | **0.5165** | 1.97 | 0.9547 |
| (c) Joint (Ours) | **0.1422** | 0.5142 | **1.95** | **0.9578** |

(b) Guidance.

| (M, N) | PSNR | LPIPS | CLIP-IQA | $E^*_{warp}$ |
|---|---|---|---|---|
| (a) $(1, 1)$ | 23.79 | 0.3242 | 0.4755 | 2.53 |
| (b) $(5, 5)$ | **24.90** | **0.2963** | 0.4931 | **1.89** |
| (c) $(\infty, 3)$ | 24.73 | 0.2984 | 0.5084 | 2.01 |
| (d) $(3, 3)$ (Ours) | 24.86 | 0.2972 | **0.5142** | 1.95 |

(c) AR chunk settings.

| Training | PSNR | LPIPS | CLIP-IQA | $E^*_{warp}$ |
|---|---|---|---|---|
| (a) w/o Patch | 24.52 | 0.3242 | 0.3997 | 2.03 |
| (b) w/o Stage-I | 24.77 | 0.3125 | 0.3980 | 1.98 |
| (c) Ours | **24.86** | **0.2972** | **0.5142** | 1.95 |

(d) Other training settings.

| DMD | PSNR | CLIP-IQA | DOVER | $E^*_{warp}$ | SC | BC |
|---|---|---|---|---|---|---|
| (a) w/o DMD | **25.04** | 0.5028 | 0.7603 | **1.87** | 0.9608 | 0.9483 |
| (b) w/ DMD (Ours) | 24.86 | **0.5142** | **0.7826** | 1.95 | **0.9632** | **0.9523** |

(e) DMD loss.

Table 4: Ablation study (a–e).

$E^*_{warp}$ and semantic consistency metrics like SC and BC. Our joint guidance strategy ensures global semantic alignment, resulting in improved consistency and enhanced perceptual fidelity.

**Influence of Chunk and Cache Size.** In Tab. 4c, we explore different chunk settings (M, N) in AR inference, where M refers to KV-cache length and N refers to the chunk length of latent frames. Using very short chunks like (1, 1) hinders the model from capturing the temporal priors of pretrained T2V models, leading to lower overall performance. Increasing the chunk size to (5, 5) brings limited gains over (3, 3) but will nearly triple the quadratic cost of DiT. Moreover, keeping the full KV cache leads to varying cache lengths at each inference step, making it harder for the model to generalize. Our default choice (3, 3) offers the best trade-off between performance and efficiency.

**Effectiveness of training settings.** We ablate key training design choices in Tab. 4d. Without Stage-I pretraining on high-resolution patches, the model struggles to adapt to long VSR sequences and shows degraded performance. Similarly, removing our proposed pixel-level window loss and training on small datasets, which enables full decoding, also hurts visual quality. This suggests that DiT backbones need proper adaptation to long-sequence VSR.

**Role of DMD loss.** As shown in Tab. 4e, the introduction of DMD leads to improvements in perceptual quality and semantic consistency. Although the pixel-wise loss alone already enables the model to utilize the pretrained DiT's parameter priors to some extent, adding DMD further improves the results by explicitly encouraging perceptual and semantic alignment. This suggests that DMD helps the model better capture the semantic priors of video diffusion.

## 5 CONCLUSION

In this work, we propose InfVSR, a scalable and efficient framework for VSR on unbounded-length sequences. By reformulating VSR as an AR-OSD paradigm, our method breaks the length limitations of existing full-sequence approaches. Through causal DiT adaptation with dual-timescale designs and single-step distillation with pixel and distribution matching losses, we enable ultra-efficient inference while preserving temporal coherence. To support evaluation on long videos, we introduce a dedicated benchmark and adopt semantic-level consistency metrics. Extensive experiments demonstrate that our method not only achieves SOTA quality, but also delivers up to 58× speed-up compared to prior multi-step methods such as MGLD-VSR. We believe InfVSR opens new possibilities for long-range video enhancement and lays the foundation for practical deployment of generic VSR systems.

## A    ETHICS STATEMENT

The research conducted in the paper conforms, in every respect, with the ICLR Code of Ethics.

## B    REPRODUCIBILITY STATEMENT

We have provided implementation details in Sec. 4. We will also release all the code and models.

## C    LLM USAGE STATEMENT

Large Language Models (LLMs) were used solely for polishing writing. They did not contribute to the research content or scientific findings of this work.

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
