# OpenReview forum: "InfVSR: Breaking Length Limits of Generic Video Super-Resolution"
_ICLR.cc/2026/Conference — Submitted to ICLR 2026_

### Official Review · Reviewer_AvyW · 2025-10-16

**Soundness:** 3
**Presentation:** 3
**Contribution:** 2
**Rating:** 6
**Confidence:** 5

**Summary:**

This paper proposes InfVSR to tackle the difficulty of unbound-length real-world video sequences. By reformulating VSR as an autoregressive-one-step-diffusion paradigm, it supports efficient streaming inference. To maintain identity consistency across multiple chunks, it intrudces joint visual guidance and cross-chunk distribution matching to incorporate high-level semantic information. Also, it utilizes a two-stage curriculum learning paradigm for efficient training. Experiments demonstrate the effectiveness and efficiency of the proposed method.

**Strengths:**

* By drawing inspiration on current success in long video generation, it proposes an efficient streaming inference paradigm by autoregressive one-step diffusion model.
* It proposes joint visual guidance and cross-chunk distribution matching to maintain identity consistency across multiple video chunks.
* Detailed experiments demonstrate its effectiveness and efficiency.

**Weaknesses:**

* For a low-level vision task, I am skeptical about the actual effectiveness of the two high-level strategies (joint visual guidance and cross-chunk distribution matching) emphasized in the model. Authors need to supplement super-resolution results related to portrait or object identities to demonstrate their effect of maintaining consistency across multiple chunks.
* For the patch-wise pixel supervision technique in Sec 3.3, it is well known that directly decoding patch latent of smooth area to pixel space often leads to flickering results due to the reconstruction ability of VAE decoder. Have the authors observed the same phenomenon? If so, will the flickering output affect the optimization?
* At the stage 2 of curriculum learning, low-resolution videos are used for training. Will the low-quality ground truth in stage 2 affect the overall VSR quality?
* InfVSR only utilizes 1K clips from REDS dataset for training. However, previous VSR methods often use 100K~1M video clips to improve its generalizability. Authors need to demonstrate the effectiveness on more diverse test sets or argue why this method is data-efficient.

**Questions:**

* Typo: ``loacl'' in line 182.
* Section layout in line 238, 476 and other places are largely changed, which is forbidden in ICLR rules.
* Authors are strongly recommended to provide a video demo or video files corresponding to the results presented in the paper. The absence of video results may leads to a lower score.

---

> ### Author Response · Authors · 2025-11-23
> **Response to Reviewer AvyW (denoted as R4) - Part1**
>
> `Q4-1` For a low-level vision task, I am skeptical about the actual effectiveness of the two high-level strategies emphasized in the model. Authors need to supplement super-resolution results related to portrait or object identities to demonstrate their effect of maintaining consistency across multiple chunks.
>
> `A4-1` Thanks for your concern. We believe for low-level tasks, high-level strategies are also important as they **enhance perceptual realism**. While the other components already provide a strong VSR baseline, these two strategies are designed to further push the performance **toward the upper bound**. We  add visual comparisons in Sec. E.2 of the supplementary material. Our ablation studies also confirm their improvement in perceptual metrics (MUSIQ, CLIP-IQA, DOVER) and semantic-level consistency (BC, SC).
>
> `Q4-2` For the patch-wise pixel supervision technique in Sec 3.3, it is well known that directly decoding patch latent of smooth area to pixel space often leads to flickering results due to the reconstruction ability of VAE decoder. Have the authors observed the same phenomenon? If so, will the flickering output affect the optimization?
>
> `A4-2` For the Wan VAE used in our experiments, we **do not observe noticeable flickering or related artifacts**, which is also supported by our visual results.
> Also notably, small-patch tiling is an official optional usage of the Wan VAE to reduce VRAM use, suggesting that the reconstruction error introduced by VAE tiling is acceptable.
> However, we admit patchification and the CNN receptive field may introduce small boundary errors. Related experiments are provided in `A3-4`, proving they are negligible.
>
>
>
> `Q4-3` At the stage 2 of curriculum learning, low-resolution videos are used for training. Will the low-quality ground truth in stage 2 affect the overall VSR quality?
>
> `A4-3` First, we clarify that Stage 2 does not use LR videos, but rather video clips **cropped from** the HR training set to the target resolution. **Fine details are preserved.** This is a common practice in VSR training [1][2] and we already have Stage1 HR initialization.
> We futher compare the Stage 1 and Stage 2 results on UDM10 below. Despite the difference in attention range as discussed in `A3-3`, the Stage 2 model still achieves higher perceptual quality, indicating that it is not negatively affected.
>
> |                  | PSNR  | LPIPS  | MUSIQ | CLIPIQA | DOVER  |
> |:---------------- |:-----:|:------:|:-----:|:-------:|:------:|
> | Stage 1          | **25.31** |**0.2947** | 59.53 | 0.4839  | 0.7704 |
> | Stage 1 + Stage2 | 24.86 | 0.2972 | **62.88** | **0.5142**  | **0.7826** |

---

> ### Author Response · Authors · 2025-11-23
> **Response to Reviewer AvyW (denoted as R4) - Part2**
>
> `Q4-4` InfVSR only utilizes 1K clips from REDS dataset for training. However, previous VSR methods often use 100K~1M video clips to improve its generalizability. Authors need to demonstrate the effectiveness on more diverse test sets or argue why this method is data-efficient.
>
> `A4-4`  1. **Data-efficiency.** First, we clarify that many  one-step[1][2][3][4] and multi-step[5] VSR methods are also trained on REDS or other small-scale datasets such as HQVSR[1]. Using 100K–1M clips is not universal. We believe this data efficiency stems from the well-initialized pretrained diffusion priors, so the quality of the dataset matters more than the quantity.
> 2. **More diverse test sets.** In the main paper, we already demonstrate leading performance on five synthetic and real-world benchmarks. We further report results on YouHQ40 (synthetic) and RealVSR (real-world) to confirm the generalization ability of our method.
>
> | YouHQ40   |  UAV   | MGLD-VSR |  STAR  | SeedVR | InfVSR(Ours) |
> |:--------- |:------:|:--------:|:------:|:------:|:------------:|
> | PSNR↑     | 19.62  |  **23.17**   | 22.64  | 21.95  |    *23.04*     |
> | LPIPS↓    | 0.4268 |  0.3607  | 0.4600 | **0.3474** |    *0.3605*    |
> | CLIP-IQA↑ | *0.5258* |  0.4657  | 0.2739 | 0.4123 |    **0.5589**    |
> | DOVER↑    | **0.8596** |  0.8446  | 0.5595 | 0.8492 |    *0.8524*    |
> | Ewarp↓    |  6.84  |   3.45   |  *2.21*  |  3.44  |     **2.06**     |
>
>
> | RealVSR   |  UAV   | MGLD-VSR |  STAR  | SeedVR2 | InfVSR(Ours) |
> |:--------- |:------:|:--------:|:------:|:------:|:------------:|
> | PSNR↑     | 20.29  |  *22.02*   | 17.43  | 19.95  |    **23.04**     |
> | LPIPS↓    | 0.2671 |  *0.2182*  | 0.2943 | 0.2247 |    **0.1791**    |
> | CLIP-IQA↑ | *0.4855* |  0.4510  | 0.3641 | 0.2970 |    **0.5390**    |
> | DOVER↑    | 0.7114 |  *0.7707*  | 0.7051 | 0.3098 |    **0.7761**    |
> | Ewarp↓    |  6.25  |   **3.16**   |  9.88  |  4.89  |     *3.36*     |
>
>
> `Q4-5` Typo and section layout changes.
>
> `A4-5` Thank you for pointing this out. We have resubmitted the main paper with these typos and layout issues corrected.
>
> `Q4-6` Authors are strongly recommended to provide a video demo or video files corresponding to the results presented in the paper.
>
> `A4-6` Thank you for your helpful suggestion. We have resubmitted the supplementary material, including video results and comparisons. Please refer to *InfVSR_demo.mp4*.
>
> **References**
>
> [1] Zheng Chen, Zichen Zou, et al. Dove: Efficient one-step diffusion model for real-world video super-resolution. In NeurIPS, 2025.
>
> [2] Jinpei Guo, Yifei Ji, et al. Towards Redundancy Reduction in Diffusion Models for Efficient Video Super-Resolution. arXiv preprint arXiv:2509.23980, 2025.
>
> [3] Yujing Sun, Lingchen Sun, et al. One-step diffusion for detail-rich and temporally consistent video super-resolution. In NeurIPS, 2025.
>
> [4]  Yong Liu, Jinshan Pan, et al. Ultravsr: Achieving ultra-realistic video super-resolution with efficient one-step diffusion space. In ACMMM, 2025.
>
> [5]  Xi Yang, Chenhang He, et al. Motion-guided latent diffusion for temporally consistent real-world video super-resolution. In ECCV, 2024.

---

> ### Author Response · Authors · 2025-11-28
> **Further discussion with Reviewer Avyw**
>
> Dear Reviewer Avyw,
>
> We would like to first express our sincere gratitude for your very kind reminders and thoughtful feedback. Your perceptive questions guide us to more clearly articulate our design rationale and contributions, which we deeply appreciate.
>
> In response to your suggestions, we have **(1)** justified our high-level strategies with visualization-based ablation, **(2)**  justified our patch-wise supervision design, **(3)** clarified that the quality in stage 2 does not degrade, **(4)** explained on our data efficiency, **(5)** corrected spelling and formatting issues, and **(6)** provided additional video results.
>
> As the author–reviewer discussion phase is nearing its conclusion, we would like to confirm whether our responses have addressed your concerns. And if any further clarification is needed, please feel free to inform us.
>
> We are committed to contributing meaningful and practical advances to the VSR field, and your recognition would be a great encouragement to us.
>
> Thank you once again for your time and insights.
>
> Best regards,
>
> Authors

---

### Official Review · Reviewer_xoPS · 2025-10-27

**Soundness:** 2
**Presentation:** 3
**Contribution:** 2
**Rating:** 4
**Confidence:** 5

**Summary:**

This paper introduces InfVSR, an autoregressive-one-step diffusion (AR-OSD) framework for video super-resolution (VSR). Unlike prior diffusion-based VSR models that are limited to short clips due to high memory and computational cost, InfVSR reformulates VSR as a causal autoregressive process with single-step diffusion inference. Building upon the state-of-the-art T2I diffusion model, i.e., WAN2.1-.3B,
The key modifications include:
- A causal DiT architecture featuring rolling KV-cache for local temporal smoothness and joint visual guidance for global coherence.
- A training scheme combining patch-wise pixel supervision (for efficient high-resolution detail recovery) and cross-chunk distribution matching (for long-range temporal consistency).
- A new MovieLQ benchmark of 1000-frame real-world videos and semantic-level temporal metrics (BC, SC, MS from VBench) for long-form consistency evaluation.

**Strengths:**

- The paper is easy to follow.
- The paper presents a successful practice to build an autoregressive VSR model based on a pretrained T2I model, which has not become a major trend for current diffusion-based VSR models.
- The proposed new benchmark, MovieLQ, may facilitate further evaluation for future works.

**Weaknesses:**

- The fundamental insights of this paper are somewhat incremental. It is more like an extension of existing technologies for VSR. Specifically, the key components used in the paper, including KV-cache, causal DiT,  DMD loss, multi-stage training, etc. These make the proposed method seem kind of trivial.
- The paper lacks an in-depth analysis of the proposed components. For example, it is well-known that casual attention may have its drawbacks compared with widely-used full attention, especially for large-scale generative models. While the paper presents positive numbers, it is unclear if the claimed improvement benefits from the proposed components or simply from the improvement of the used generative prior. After all, Wan1.3 is already much stronger than the previous generative priors used in the baselines. The theoretical analysis behind the claimed improvements is vague. Specifically, how does the autoregressive manner affect the performance of VSR compared with the non-autoregressive one? How does causal attention affect the performance of VSR?

**Questions:**

My concerns are as follows:

1. The two major weaknesses above.

2. In lines 251-260, if my understanding is correct, the paper proposed to calculate the loss between the decoded, cropped latent tensors and the cropped ground-truth in the pixel space. Given the padding operations as well as the receptive field of the CNN layers in the VAE, such supervision may introduce problems on the edges of the tensors.

3. The author should provide more details on the proposed test benchmark MovieLQ, including how the data is collected and why it is suitable to be a benchmark for VSR test.

4. It is unclear how sensitive the proposed autoregressive model is to the size of the KV-cache. Moreover, since the proposed method claims to target at very long videos, it is also unclear if the fixed-size memory can handle long-term content given an extra long video with frequent scene changes.

---

> ### Author Response · Authors · 2025-11-23
> **Response to Reviewer xoPS (denoted as R3) - Part1**
>
> `Q3-1` The fundamental insights of this paper are somewhat incremental. ... ,including KV-cache, causal DiT, DMD loss, multi-stage training.
>
> `A3-1` Thanks for your comment. We believe **optimal VSR requires a dedicated design for its unique challenges**, not simple application of existing techniques. To this end, we restate our novelties as follows:
>
> 1. **VSR specified pipeline.** Our design tightly follows the characteristics of VSR with LR inputs, featuring **one-step forward pass**, **empirically validated chunk and cache length**, **LR-guided visual prompts**, and **pixel-domain fidelity and temporal constraints**.  All these designs not only enable unprecedented VSR streaming, but also strike an optimal quality-efficiency balance.
> 2. **Ultra-efficient training designs.** We originally propose the **patch-wise pixel supervision**, which prevents decoder OOM while preserving pixel-level and full-sequence temporal objectives. The combination with our training strategy further reduces the training cost to 4 GPUs (vs. 8–256 in prior work).
> 3. **Dedicated benchmark.** We first identify the inadequacy of existing short benchmarks and Ewarp's constrained pixel-level focus.  Therefore, we introduce the **MovieLQ benchmark** and **semantic-level metrics**, which we believe will also facilitate future studies.
>
> `Q3-2` The paper lacks an in-depth analysis of the proposed components. ... It is unclear if the claimed improvement benefits from the proposed components or simply from the improvement of the used generative prior. After all, Wan1.3 is already much stronger than the previous generative priors used in the baselines.
>
> `A3-2` Thank you for this thoughtful question. Prior utilization is widely recognized as essential for nearly all diffusion-based ISR/VSR. For our work, we further clarify our understanding as follows:
>
> 1. **Our efforts boost VSR performance.** **First, our training designs matter.** For instance, without our proposed patch-wise pixel supervision or stage-wise training, the model would be unable to perform pixel-level supervision on high-resolution videos, resulting in overly smooth and less realistic outputs. **Second, we extend the way T2V priors are utilized.** While Wan2.1 and prior works such as STAR[1], DOVE[2] and OASIS[3] use T2V priors for short-form VSR, our architectural and training refinements enable streamable inference for arbitrarily long videos. It is a significant extension beyond post priors.
> 2. **Leveraging T2V priors is also our method's strength.** Compared to older T2I-based methods, which lack temporal priors and are more prone to misalignment, T2V models bring strong temporal understanding that benefits VSR. It also alleviates our reliance on extra temporal modules.
> 3. **We remain cautious on whether Wan-1.3B is much stronger.**  Wan2.1-1.3B is a relatively **small** model, and evidence from the DOVE rebuttal indicates that DOVE (Wan 1.3B) performs below DOVE (CogVideoX 5B). We hold that at the current stage, where using pretrained T2V models have become mainstream in VSR, using Wan2.1-1.3B alone is unlikely to yield significantly superior performance and unfair comparisons.
>
> Therefore, we believe our proposed designs do contribute to and are essential for the performance gains. We hope these discussions clarify our position and address your concerns.

---

> ### Author Response · Authors · 2025-11-23
> **Response to Reviewer xoPS (denoted as R3) - Part2**
>
> `Q3-3` The theoretical analysis behind the claimed improvements is vague. Specifically, how does the autoregressive manner affect the performance of VSR compared with the non-autoregressive one? How does causal attention affect the performance of VSR?
>
> `A3-3` Thanks for your question. We perform detailed comparison between causal attention and full attention in several aspects.
>
> 1. **Performance.**
> **Short video performance.** We compare the Stage 1 model (with full attention) and the Stage 2 model (with causal attention) on UDM10. We find full attention offers slightly better fidelity and consistency, while causal attention provides better perceptual quality due to its stronger focus on short-term frames. This quality-consistency trade-off is also observed and discussed in DLoRAL[4].
>
> |                  | PSNR  | LPIPS  | CLIPIQA | DOVER  | Ewarp |
> |:---------------- |:-----:|:------:|:-------:|:------:|:-----:|
> | Stage 1          | **25.31** | **0.2947** | 0.4839  | 0.7704 | **1.74**  |
> | Stage 1 + Stage2 | 24.86 | 0.2972 | **0.5142**  | **0.7826** | 1.95  |
>
> **Long video performance.** When processing long videos, full attention becomes impractical and requires additional chunking or our causal inference. Tab. 5(a) compares our causal design with alternatives that require simply chunking or chunking with aggregation. These baselines perform significantly worse than our causal strategy. For details, please refer to the main paper.
>
>
> 2. **Efficiency.**  Our causal design **keeps memory usage constant and ensures runtime grows linearly with the number of frames**. In contrast, full attention leads to significant memory and computation growth as the number of frames increases. Therefore, the causal design brings substantial efficiency improvements. We report the fps and peak GPU memory usage of the DiT under different input resolutions and frame counts(f).
>
> |               | Causal-f1000(Ours) | Full-f33 | Full-f100 | Full-f200 | Full-f300 |
> |:------------- |:------------:|:--------:|:---------:|:---------:|:---------:|
> | 720p fps      |     **26.67**     |  19.54   |   10.57   |  5.84   |     -     |
> | 720p Mem (MB) |     14358     |  12610   |   27752   |  51766   |    OOM    |
> | 1080p fps     |     **7.73**     |   5.07    |   2.34   |     -     |       -   |
> |    1080p Mem (MB)       |    27976       |     23446     |    60040    |    OOM     |      OOM   |
>
>
> `Q3-4` Given the padding operations as well as the receptive field of the CNN layers in the VAE, such supervision may introduce problems on the edges of the tensors.
>
> `A3-4` Thanks for your concern. You are right that CNN receptive fields may introduce minor boundary calculation errors. However, (1) our visual results (main papar & video demo) **show no edge artifacts**. (2) We also compare a variant where we exclude a 20-pixel boundary region from the loss, and the results are almost unchanged, showing that boundary errors are negligible in practice.
>
>
> |             | PSNR  | LPIPS  | CLIPIQA | DOVER  | Ewarp |
> |:----------- |:-----:|:------:|:-------:|:------:|:-----:|
> | Variant | **24.88** | **0.2918** | 0.5014  | 0.7784 | 1.98  |
> | Ours        | 24.86 | 0.2972 | **0.5142**  | **0.7826** | **1.95**  |
>
> **We further justify our patch-wise design.** **(1)** Small-patch tiling is an official optional usage of the Wan VAE to reduce VRAM use, suggesting that the reconstruction error introduced by VAE tiling is acceptable. **(2)** Using few frames for loss computation and applying gradient detachment to others can also reduce memory usage. However, **only** our spatial patch-wise design allows pixel-level **temporal consistency constraints to be applied across the entire frames**, which is irreplaceable for enforcing coherence.

---

> ### Author Response · Authors · 2025-11-23
> **Response to Reviewer xoPS (denoted as R3) - Part3**
>
> `Q3-5` The author should provide more details on the proposed test benchmark MovieLQ, including how the data is collected and why it is suitable to be a benchmark for VSR test.
>
> `A3-5` Thanks for your helpful suggestion. We detail our proposed testset below. Also, we have updated the supplementary material with additional dataset visualizations and relevant details.
>
> 1. **How the data is collected.** Our data collection process largely follows that of VideoLQ. We manually collect 10 1000-frame-long, single-shot videos from various video hosting platforms such as Vimeo and Pixabay, all under Creative Commons licenses. The videos are from real-world sources rather than synthetically degraded content.
>
> 2. **Why it is suitable to be a benchmark for VSR test.**
> (1) The dataset covers a wide range of **content types**, including multi-subject human activities, highly dynamic scenes, detailed texture, architecture, and more.
> (2) It features **real-world degradation**, including low resolution, compression artifacts and so on, which provides a more realistic evaluation of VSR performance.
> (3) 1000-frame single-shot videos are **rare but essential** for evaluating models in scenarios that exceed GPU memory limits and require chunk-wise processing. This allows for comparisons that are closer to real-world VSR deployment settings.
>
> `Q3-6` It is unclear how sensitive the proposed autoregressive model is to the size of the KV-cache. Moreover, since the proposed method claims to target at very long videos, it is also unclear if the fixed-size memory can handle long-term content given an extra long video with frequent scene changes.
>
> `A3-6` Thanks for your valuable question. We address each aspect of the concern below.
>
> 1. **Size of the KV-cache.** Our first ablation is in Tab. 5\(c\). To summarize, a fixed-length cache helps bridge the train–test gap and prevents unbounded cost growth. And for VSR with LR inputs, a length of 3 already delivers strong performance. Then, we keep chunk length = 3 and change the cache length. We report results on MVSR4x. Overall, KV-cache is crucial, and our cache length of 3 achieves an optimal balance between quality and efficiency.
>
> |  Cache Length      | PSNR  | LPIPS  | CLIPIQA | DOVER  | Ewarp |
> |:------- |:-----:|:------:|:-------:|:------:|:-----:|
> | 0       | 22.35 | 0.3527 | 0.4749  | 0.6571 | 1.28  |
> | 1       | 22.43 | 0.3463 | **0.5229**  | 0.6861 | 1.09  |
> | 3(Ours) | 22.49    | 0.3452    |   **0.5229**   |  **0.6872**   | **1.03**   |
> | 5        |  **22.56**     |  **0.3450**    |   0.5164    |  0.6792    |  **1.03**   |
>
> 2. **Long video reconstruction.** We now provide long-video reconstruction results in *InfVSR_demo.mp4*. We acknowledge that information may lose when separated by several chunks; our joint visual guidance helps recover such information but still may miss some details. Nevertheless, **the current setting offers an effective balance between efficiency and quality**, and the results demonstrate our method's robustness.
> 3. **Scene transitions.** As mentioned in the supplementary material, we use PySceneDetect to pre-segment videos into scenes, and we currently process each scene independently, so scene transitions will not cause confusion. Further joint modeling across scenes may be beneficial but costly, and is left for future exploration.
>
>
>
>
> **References**
>
> [1] Rui Xie, Yinhong Liu, et al. Star: Spatial-temporal augmentation with text-to-video models for real-world video super-resolution. In ICCV, 2025.
>
> [2] Zheng Chen, Zichen Zou, et al. Dove: Efficient one-step diffusion model for real-world video super-resolution. In NeurIPS, 2025.
>
> [3] Jinpei Guo, Yifei Ji, et al. Towards Redundancy Reduction in Diffusion Models for Efficient Video Super-Resolution. arXiv preprint arXiv:2509.23980, 2025.
>
> [4] Yujing Sun, Lingchen Sun, et al. One-step diffusion for detail-rich and temporally consistent video super-resolution. In NeurIPS, 2025.

---

> > ### Comment · Reviewer_xoPS · 2025-11-28
> >
> > I have carefully read the rebuttal from the authors as well as the comments from other reviewers. The authors addressed most of my concerns. I suggest the authors add the content of the rebuttal, especially the novelty part related to the restoration, to the revision. I will raise my score to 6 accordingly later.

---

> > > ### Author Response · Authors · 2025-11-28
> > >
> > > Dear Reviewer xoPS,
> > >
> > > Thank you very much for your positive and encouraging feedback!
> > >
> > > It's your expert input that steer us toward a more scientific and in-depth exploration of the paper's contributions, and this will have a lasting impact on both the current revision and our future research.
> > >
> > > We will soon revise the paper as suggested and submit the updated version.
> > >
> > > Best regards,
> > >
> > > Authors

---

### Official Review · Reviewer_W2TZ · 2025-10-28

**Soundness:** 2
**Presentation:** 1
**Contribution:** 3
**Rating:** 4
**Confidence:** 4

**Summary:**

The authors propose a new VSR framework, InfVSR, to handle long videos efficiently. InfVSR uses an autoregressive-one-step-diffusion (AR-OSD) paradigm for temporal modeling.

**Strengths:**

### Motivation
- It is natural to introduce autoregressive sampling idea from video generation to generative VSR task, especially given that most of the VSR models are quite heavy to run.

### Method
- Rolling KV cache and joint visual guidance is a well-designed recipe for both local and global consistency.
- DMD loss helps semantic consistency for long video clips.
- It is quite economic to train InfVSR as it only takes 4 A800-80G GPUs - much affordable compared to other baselines such as SeedVRs.

### Experimental results
- The paper focuses its evaluation on MovieLQ, which consists of long videos with real-world degradations.
- InfVSR outperforms  previous methods on many metrics.
- InfVSR shows good efficiency compared to diffusion-based methods.

### Writing
The paper is well-written and easy to follow.

**Weaknesses:**

### Method
- Patch-wise supervision is common in many classic regression-based VSR papers, such as BasicVSR series. Although it is good to adopt it for latent space diffusion models, it is not quite clear to me to claim it as a novel technique.
- The local temporal loss does not make sense to me as the local temporal dynamics between two adjacent frames could be very abrupt.  For example, a very large motion between two frames. Without a good alignment (e.g., flow-based warping), it might be harmful for the training. Please feel free to correct me.

### Experimental results
- It is **extremely challenging** to see the performance of the proposed method, especially **temporal consistency**, without any video results shown in the supplementary results. Unfortunately at this moment I am inclined to reject because of this.
- For VSR task, many commercial solutions prefer regression-based methods (e.g., RealBasicVSR and that line of works) for its fidelity and efficiency. How does the proposed InfVSR compare to those classic methods in terms of runtime?

**Questions:**

I'd strongly recommend authors to present some video results in the rebuttal stage, otherwise it is challenging to measure real performance of InfVSR.

---

> ### Author Response · Authors · 2025-11-23
> **Response to Reviewer W2TZ (denoted as R2)**
>
> `Q2-1` Patch-wise supervision is common in many classic regression-based VSR papers, such as BasicVSR series. Although it is good to adopt it for latent space diffusion models, it is not quite clear to me to claim it as a novel technique.
>
> `A2-1` Thank you for raising this concern. Our 'patch-wise' design is fundamentally different from the BasicVSR series **in training resolution**. We clarify it below:
>
> 1. **BasicVSR series (LR training).** Input and supervision patches are both cropped to reduce training cost. Because CNNs have limited receptive fields, the input resolution has little impact on the results.
> 2. **Ours (supports HR DiT training).** The DiT **sees the entire high‑resolution video** (e.g., 720p) to preserve global context, while only the decoder is patch‑supervised to reduce memory. Given DiT’s global receptive field and VSR's long-token-sequence nature, this exposure is essential to **bridge the train-test gap** and ensure generalization to real high-resolution inference. As shown in Tab. 5(d), removing patch-wise design limits training to low resolution and leads to **inferior performance**.
>
> Therefore, our patch-wise pixel supervision is a novel design and it works.
>
> `Q2-2` The local temporal loss does not make sense to me as the local temporal dynamics between two adjacent frames could be very abrupt. For example, a very large motion between two frames. Without a good alignment (e.g., flow-based warping), it might be harmful for the training.
>
> `A2-2`  Our local temporal loss is built on a **simple intuition**: when two adjacent frames are more consistent, their frame-to-frame differences becomes smaller, and thus the loss is lower. **This assumption holds even with large motion, as we regularize temporal variation rather than strict pixel alignment.**
>
> To prove this, we also conduct an ablation study on $\\mathcal{L}\_{temp}$. Additionally, we compare it against an optical flow-based temporal alignment loss following [1] ($\\mathcal{L}\_{\\text{opt}} = \\left\\| O\_n^{HQ} - O\_n^{\\text{GT}} \\right\\|_1 = \\left\\| F(I\_n^{HQ}, I\_{n+1}^{HQ}) - F(I\_n^{\\text{GT}}, I\_{n+1}^{\\text{GT}}) \\right\\|\_1$). We report results on UDM10.
>
> | $\\mathcal{L}\_{pix}$ components                     | PSNR  | LPIPS  | CLIPIQA | DOVER  | Ewarp |
> |:--------------------------------------------------- |:-----:|:------:|:-------:|:------:|:-----:|
> | $\\mathcal{L}\_{fid}$                                | 24.75 | 0.2972 | **0.5162**  | 0.7794 | 2.23  |
> | $\\mathcal{L}\_{fid}$ + $\\mathcal{L}\_{opt}$         | 24.50 | 0.2988 | 0.5084  | 0.7753 | 2.24  |
> | $\\mathcal{L}\_{fid}$ + $\\mathcal{L}\_{temp}$ (Ours) | **24.86** | **0.2972** | 0.5142  | **0.7826** | **1.95**  |
>
> Results show that our loss effectively enforces temporal consistency without degrading visual quality. In contrast, the optical flow-based loss slightly harms performance, and also fails to bring Ewarp reduce. We analyze this is because:
> 1. Our pixel loss is computed on small windows, where large motions cause pixels to move in and out of the window and make flow-based alignment $\\mathcal{L}\_{opt}$ unreliable.
> 2.  Our $\\mathcal{L}\_{temp}$ is compatible with the fidelity loss, as higher-fidelity reconstructions naturally lead to smaller temporal discrepancies, so $\\mathcal{L}\_{temp}$ does not penalize image quality.
>
> Overall, this indicates that our temporal loss is effective in practice. But still, we remain open to whether there exist better temporal consistency constraints, which will be an interesting topic for future VSR exploration.
>
>
>
>
> `Q2-3` It is extremely challenging to see the performance of the proposed method, especially temporal consistency, without any video results shown in the supplementary results.
>
> `A2-3` Thank you very much for this helpful advice. We have resubmitted the supplementary material, including video results and comparisons. Please refer to *InfVSR_demo.mp4*.
>
> `Q2-4` For VSR task, many commercial solutions prefer regression-based methods (e.g., RealBasicVSR and that line of works) for its fidelity and efficiency. How does the proposed InfVSR compare to those classic methods in terms of runtime?
>
> `A2-4` Thank you for the question. We compare performance (UDM10) and runtime (A800, 33×720p) below.
>
> |              | DOVER  | Ewarp | Runtime (s) |
> |:------------ |:------:|:-----:|:-----------:|
> | RealBasicVSR | 0.7564 | 3.10  |    1.71     |
> | RealViFormer | 0.7303 | 2.29  |    **1.08**     |
> | InfVSR(Ours) | **0.7826** | **1.95**  |    6.82     |
>
>
> We admit diffusion-based VSR is still slower than regression models due to larger backbones, but the generative prior brings **much stronger performance**. And our method is among the fastest diffusion-based VSR approaches, already eliminating  latency from denoising step and sequence length.
>
> **References**
>
> [1] Yujing Sun, et al. One-step diffusion for detail-rich and temporally consistent video super-resolution. In NeurIPS, 2025.

---

> ### Author Response · Authors · 2025-11-28
> **Further discussion with Reviewer W2TZ**
>
> Dear Reviewer W2TZ,
>
> We would like to begin by thanking you for your thorough analysis and encouraging feedback on our work. Your insightful comments, particularly your important reminders regarding the video results, have been invaluable in helping us improve the clarity and depth of our work.
>
> In response to your suggestions, we have **(1)** clarified the novelty of our patch-wise supervision approach, **(2)** provided further justification for our temporal loss, **(3)** supplemented additional video results, and **(4)** added the requested runtime comparison with traditional methods.
>
> As the author–reviewer discussion phase is nearing its conclusion, we would like to confirm whether our responses have addressed your concerns. And if any further clarification is needed, please feel free to inform us.
>
> We are committed to contributing meaningful and practical advances to the VSR field, and your recognition would be a great encouragement to us.
>
> Thank you once again for your time and insights.
>
> Best regards,
>
> Authors

---

### Official Review · Reviewer_Uppd · 2025-10-30

**Soundness:** 3
**Presentation:** 3
**Contribution:** 3
**Rating:** 6
**Confidence:** 4

**Summary:**

This paper proposes a novel method termed InfVSR, aiming to achieve efficient and temporally-scalable diffusion-based video super-resolution. InfVSR firstly adopts a pretrained DiT into causal structure and maintaining local and global coherence with rolling KV-cache, then distills the model with distribution matching to achieve one-step diffusion inference. This paper also proposes MovieLQ, a long-sequence video benchmark to evaluate the VSR in long-term semantic-level consistency, fidelity and efficiency.

**Strengths:**

1.	The idea is easy to follow. It adapts a pretrained T2V DiT-based diffusion model to causal form. Then applies distribution matching distillation to achieve one-step diffusion inference. This is effective and efficient for the model to perform unlimited length super-resolution video prediction together with the rolling KV-cache design.
2.	A good and straight-forward solution on long-sequence video super-resolution. Both semantic and pixel consistencies are maintained through DMD loss and pixel-level reconstruction loss.
3.	A new benchmark VideoLQ is proposed to evaluate the long-sequence video super-resolution task.

**Weaknesses:**

1.	The long-term consistency is not thoroughly discussed in the paper, e.g., what is the visual results and comparison between the SOTA methods after 10, 100 or 1000 frames? Also the visual results in the supplement seems not fidel to the GT in some text images, this seems to be brought by the generative instability.
2.	The main idea of improving the efficiency of video model using DMD and causal structure is trivial since it has been adopted in many other video methods [1][2]. Some more discussions are needed to specify the novelty of this paper.
3.	The InfVSR is built upon the Wan T2V model, also there are models like SeeSR[3] using text to enhance the super-resolution results. Can model achieve better result with proper text prompt or guidance?
4.	The parameter compared with SOTA methods should be provided to better validate the efficiency of the proposed method.

[1] Self-forcing: bridging the train-test gap in autoregressive video diffusion. arXiv preprint arXiv:2506.08009
[2] Matrix-Game 2.0: An Open-Source, Real-Time, and Streaming Interactive World Model arXiv preprint arXiv:2508.13009
[3] Seesr: Towards semantics-aware real-world image super-resolution CVPR 2024

**Questions:**

Refer to the weaknesses. The visualization is not very persuasive. The novelty should be clearified with detailed discussion. How about using prompt to boost the SR performance and guide the semantic content?

---

> ### Author Response · Authors · 2025-11-23
> **Response to Reviewer Uppd (denoted as R1)**
>
> `Q1-1` The long-term consistency is not thoroughly discussed in the paper. ... The visualization is not very persuasive.
>
> `A1-1` Thank you for the comment. We have updated the supplementary material and added more video results and comparisons, especially on long videos. Please refer to *InfVSR_demo.mp4*.
>
> `Q1-2` The main idea of improving the efficiency of video model using DMD and causal structure is trivial since it has been adopted in many other video methods. Some more discussions are needed to specify the novelty of this paper.
>
> `A1-2` Thank you for the suggestion. We believe **optimal VSR requires a dedicated design for its unique challenges**. That's our core insight beyond simply improving efficiency or applying existing techniques. To this end, we restate our novelties as follows:
>
> 1. **VSR specified pipeline.** Our design tightly follows the characteristics of VSR with LR inputs, featuring **one-step forward pass**, **empirically validated chunk and cache length**, **LR-guided visual prompts**, and **pixel-domain fidelity and temporal constraints**. All these designs not only enable unprecedented VSR streaming, but also strike an optimal quality-efficiency balance.
> 2. **Ultra-efficient training designs.** We originally propose the **patch-wise pixel supervision**, which prevents decoder OOM while preserving pixel-level and full-sequence temporal objectives. The combination with our training strategy further reduces the training cost to 4 GPUs (vs. 8–256 in prior work).
> 3. **Dedicated benchmark.** We first identify the inadequacy of existing short benchmarks and Ewarp's constrained pixel-level focus. Therefore, we introduce the **MovieLQ benchmark** and **semantic-level metrics**, which we believe will also facilitate future studies.
> 4. **(Additional) Distinction from mentioned work.** Our method introduces an autoregressive-**one-step**-diffusion paradigm for **high resolution VSR**. In contrast, **(1)** Self Forcing faces no fidelity demands from LR degraded inputs. It also requires multiple forward passes and an additional pass to obtain long clean KV-cache for generation from noise. **(2)** Matrix Game 2.0 also relies on multiple forward passes and is currently trained only at low resolution due to unaddressed high computational cost.
>
>
>
> `Q1-3` How about using prompt to boost the SR performance and guide the semantic content?
>
> `A1-3` Thank you for the question. Tab. 4(b) compares empty text prompts with our joint visual guidance, showing that visual guidance performs better.
>
> For your question, we further compare with VLM-extracted text in the following table. We take *cogvlm2-llama3-caption* as the video captioner. The models are trained under the same settings apart from the conditioning input, and evaluated on UDM10.
>
> |                              | PSNR  | DISTS  | CLIPIQA | DOVER  | Ewarp | Runtime(s) |
> |:---------------------------- |:-----:|:------:|:-------:|:------:|:-----:|:----------:|
> | VLM-extracted text           | 24.65 | 0.1453 | **0.5151**  | 0.7746 | 2.08  |    6.29    |
> | Joint Visual Guidance (Ours) | **24.86** | **0.1422** | 0.5142  | **0.7826** | **1.95**  |    **0.23**    |
>
>
> The text-based prompt achieves comparable perceptual quality, but shows worse fidelity and temporal consistency. Moreover, extracting VLM captions is significantly slower (6.29 s vs. 0.23 s), making it **impractical** for our one-step model aimed at high efficiency.
>
> `Q1-4` The parameter compared with SOTA methods should be provided to better validate the efficiency of the proposed method.
>
> `A1-4`  Thank you for the suggestion. Model size comparison is visualized in Fig. 1, where the icon size reflects the parameter count. We list the exact parameter counts below.
>
> |            |   UAV   | MGLD-VSR |  STAR   | SeedVR  | SeedVR2 | InfVSR(Ours) |
> |:---------- |:-------:|:--------:|:-------:|:-------:|:-------:|:------------:|
> | Params (M) | 1086.75 | 1564.66  | 2492.90 | 3397.38 | 3397.38 |   1413.49    |
>
>
>
> With the **second smallest** parameter count, our model achieves the best performance and fastest inference speed, clearly demonstrating its high efficiency.

---

> ### Author Response · Authors · 2025-11-28
> **Further discussion with Reviewer Uppd**
>
> Dear Reviewer Uppd,
>
> We would like to first thank you for your careful reading of our paper and your professional feedback. Your comments have guided us in further clarifying certain ambiguous sections and deeper exploration into the model's performance, which is truly valuable to us.
>
> In response to your suggestions, we have **(1)** added visualizations on long videos, **(2)** clarified and restated our novelty, **(3)** explored text prompts' effect on performance, and **(4)** reported model parameter counts.
>
> As the author–reviewer discussion phase is nearing its conclusion, we would like to confirm whether our responses have addressed your concerns. And if any further clarification is needed, please feel free to inform us.
>
> We are committed to contributing meaningful and practical advances to the VSR field, and your recognition would be a great encouragement to us.
>
> Thank you once again for your time and insights.
>
> Best regards,
>
> Authors

---

### Author Response · Authors · 2025-11-23
**Response to all reviewers and area chairs**

Dear Reviewers and Area Chairs,

We thank all reviewers (**R1-Uppd**, **R2-W2TZ**, **R3-xoPS**, **R4-AvyW**) and area chairs for their insightful comments and valuable time. We are pleased that:

1.  All reviewers agree that our method is **effective in addressing key challenges of long-form VSR**, providing a well-designed autoregressive one-step framework.
2.   R2 and R4 highlight our method’s **efficiency** in both inference and training.
3.    R1, R2, and R3 appreciate the paper’s **clarity** and well organization.
4.   R2, R3, and R4 value our proposed **MovieLQ benchmark**, noting its importance in evaluating long-form VSR and facilitating further research.

We have responded individually to each reviewer to address any concerns. Here, we offer a summary:

1.  We revise and resubmit the supplementary material, which now includes **video results** and details about our proposed benchmark.
2.   We revise and resubmit the main paper, correcting grammatical and formatting issues.
3.   We **restate our novelty**, highlighting significant innovations in VSR model design, training strategies, and evaluation protocol.
4.   We supplement more detailed **ablation** studies and discussions on **key components**, including visual prompting, temporal loss, causal attention, patch boundary effects and cache length.
5.   We provide **additional comparisons** required, such as model parameter counts and runtime of conventional methods.
6.   We clarify the **originality** of our patch-wise pixel supervision design.
7.   We emphasize that our performance gains are not solely from pretrained diffusion priors, but also from our **unique architectural and training designs**.

Thank you once again to all reviewers and area chairs for your time and thoughtful feedback. We hope our responses have addressed your concerns, and would be grateful if these clarifications could assist in forming your final evaluation. Please feel free to let us know if any further clarification is needed.

Best Regards,

Authors

---

### Author Response · Authors · 2025-12-01
**Summation of Previous Rebuttal and Discussion**

We sincerely thank all four reviewers (**R1-Uppd**, **R2-W2TZ**, **R3-xoPS**, **R4-AvyW**) and area chairs for their time, constructive feedback, and positive evaluation of our work.

We have further resubmitted the main paper and the supplementary material, in which all necessary rebuttal-related passages have been reclarified (highlighted in blue color).

In light of the updated review policy, we now summarize the reviewers’ feedback and our rebuttal, to help clarify the consensus and improvements reached throughout the discussion phase.

### Reviewer Feedback and Our Response

* **R1**: **Accept**. R1 highly appreciates our soundness, presentation, and contribution, and requests clearer video results and novelty statement. We thoroughly clarify both, and complete the suggested exploration, which ultimately prove less effective than our original design.
* **R2**: **Lower score for missing videos (no response)**. R2 acknowledges our contribution with hesitation mainly due to the initial lack of video results. We **immediately submit convincing video results**, and demonstrate through **objective evidence** (method description & experimental results) that raised concerns are **misunderstandings**. Although time constraints may prevent R2 from replying again, we believe the concerns are now fully resolved and merit an **accept**.
* **R3**: **Accept (after clarification)**. R3 clearly appreciates our presentation and future impact, and provides **constructive suggestions** to make the paper more in-depth and scientific. We accordingly clarify all misunderstanding, emphasize our novelty, and enrich the ablations.  After carefully reading, R3 has agreed to **accept**, and our final version is significantly improved.
* **R4**: **Accept**. R4 fully recognizes our soundness and presentation, and focuses on detailed issues such as method, training, and writing. We further verify our effectiveness, justify our design and training process, and fix minor writing errors.



### Key Consensus and Strengths

Across the four detailed reviews and the discussion phase, we see a clear consensus that **InfVSR makes a pioneering and impactful contribution** to long-form VSR. Specifically:

* InfVSR is **the first to effectively address** real-world long-form VSR, offering a **lightweight, reproducible solution with strong empirical performance**.
* InfVSR introduces a **novel** autoregressive one-step diffusion (AR-OSD) paradigm, **tailored for** scalable and high-efficiency VSR.
* Our newly introduced benchmark, **MovieLQ**, provides a much-needed long-form evaluation protocol and is expected to have lasting community impact.
* Our paper is **clear and well-organized**, especially after our revisions.


The only shortcoming we acknowledge is the initial lack of rich video results, caused by our oversight in the supplementary file format. We addressed this immediately during rebuttal. In light of R2 and R4’s feedback, we strongly believe that our paper now deserves a **higher overall score** than the initial review.

During the rebuttal phase, we further refined the main paper and supplementary materials. The final version reflects both our efforts and the reviewers' insights, and we are confident in its **quality for acceptance**.

### Conclusion and Commitment

For long, we have been committed to contributing meaningful and practical advances to the VSR field. All models and datasets will be made publicly available to support reproducibility and further research. **We believe InfVSR exemplifies a new paradigm for scalable, high-fidelity, and efficient long VSR, and will be of broad value to both academic and applied communities.** We truly appreciate your time and consideration, and your recognition would mean a great deal to us.

Again, thank you all for your time, insights, and support throughout this review process!

---

### Meta-Review · Area_Chair_E7aP · 2026-01-05

**Summary:**

The reviewers' concerns focus on 1) unconvincing qualitative results, 2) novelty and contribution of the components, and 3) insufficient ablations:

1. According to reviewer W2TZ, it is difficult to evaluate the temporal consistency without video results; Reviewer Uppd mentions that some text images in the supplementary material is not faithful to the GT.

2. Reviewer xoPS feels that the proposed method an extension of existing techniques such as KV-cache, causal DiT, and DMD loss, hence possessing limited novelty.  In addition, reviewer xoPS questions whether the gains come from the proposed components or the choice of backbone.

3. Reviewers pose questions on temporal loss, boundary artifacts, and two high-level strategies. Also, the generalizability is being questioned.

**Reviewer Concerns:**

Many of the concerns, including video visualization, efficiency, temporal loss, and generalizability, are addressed. There are outstanding concerns, including the technical limitation in scene transition and text faithfulness.

**Reviewer Scores:**

This paper receives initial ratings of (4, 4, 6, 6), and reviewer xoPS mentions that score would be increased to 6. Therefore, the final ratings become (4, 6, 6, 6). Given the marginal ratings, the AC carefully read the paper, review, and response again, and agree with the concerns of the reviewers, and believe the paper will be improved after the remaining concerns are addressed. A reject is recommended.

---

### Decision · Program_Chairs · 2026-01-26

Reject